# Direct On-Chip Diagnostics of *Streptococcus bovis*/*Streptococcus equinus* Complex in Bovine Mastitis Using Bioinformatics-Driven Portable qPCR

**DOI:** 10.3390/biom14121624

**Published:** 2024-12-18

**Authors:** Jaewook Kim, Eiseul Kim, Seung-Min Yang, Si Hong Park, Hae-Yeong Kim

**Affiliations:** 1Institute of Life Sciences & Resources, Department of Food Science and Biotechnology, Kyung Hee University, Yongin 17104, Republic of Korea; jwkim773@gmail.com (J.K.); eskim89@khu.ac.kr (E.K.); ysm9284@gmail.com (S.-M.Y.); sihong.park@oregonstate.edu (S.H.P.); 2Department of Food Science and Technology, Oregon State University, Corvallis, OR 97331, USA

**Keywords:** *Streptococcus bovis*/*Streptococcus equinus* complex, genetic markers, portable qPCR, dairy product

## Abstract

This study introduces an innovative on-site diagnostic method for rapidly detecting the *Streptococcus bovis*/*Streptococcus equinus* complex (SBSEC), crucial for livestock health and food safety. Through a comprehensive genomic analysis of 206 genomes, this study identified genetic markers that improved classification and addressed misclassifications, particularly in genomes labeled *S. equinus* and *S. lutetiensis*. These markers were integrated into a portable quantitative polymerase chain reaction (qPCR) that can detect SBSEC species with high sensitivity (down to 10^1^ or 10^0^ colony-forming units/mL). The portable system featuring a flat chip and compact equipment allows immediate diagnosis within 30 min. The diagnostic method was validated in field conditions directly from cattle udders, farm environments, and dairy products. Among the 100 samples, 51 tested positive for bacteria associated with mastitis. The performance of this portable qPCR was comparable to laboratory methods, offering a reliable alternative to whole-genome sequencing for early detection in clinical, agricultural, and environmental settings.

## 1. Introduction

Subclinical and clinical mastitis in dairy cattle, primarily caused by pathogenic bacteria, is a significant concern in the dairy industry, leading to substantial economic losses due to reduced milk yield, altered milk composition, and increased veterinary costs [1]. Among the various pathogens associated with bovine mastitis, *Streptococcus bovis*/*Streptococcus equinus* complex (SBSEC) members have emerged as notable culprits [2]. SBSEC members include *S. equinus* (synonymous with *S. bovis*), *S. gallolyticus* subsp. *gallolyticus*, *S. gallolyticus* subsp. *pasteurianus*, *S. gallolyticus* subsp. *macedonicus*, *S. infantarius* subsp. *infantarius*, *S. lutetiensis*, *S. alactolyticus*, and *S. ruminicola*. These bacteria are Gram-positive, catalase-negative cocci widely distributed in the environment, particularly within the gastrointestinal tracts of ruminants [3]. SBSEC members are implicated in various infections in animals and humans, with bovine mastitis being a particularly significant issue in dairy cattle. SBSEC pathogenicity in mastitis is linked to their ability to adhere to and invade mammary epithelial cells, evading the host’s immune response and establishing chronic infections that are often challenging to detect and treat [4]. Interestingly, SBSEC also includes strains that play a beneficial role in the dairy industry, particularly in the fermentation of dairy products, contributing to the flavor, texture, and safety of fermented milk products [5]. This dual role of SBSEC members as pathogens in bovine mastitis and beneficial microorganisms in dairy fermentation introduces complexity in accurately identifying and distinguishing pathogenic from nonpathogenic strains in dairy contexts.

Given the significant economic and health implications of SBSEC-associated mastitis, rapid detection of these bacteria is essential for effective disease management. In the dairy industry, rapid and efficient field diagnostics are crucial for managing diseases, such as bovine mastitis, which greatly impact milk production and quality [6]. Traditional diagnostic methods, such as bacterial culture and biochemical tests, are effective, but often require time-consuming laboratory procedures and skilled personnel, making them less practical for immediate on-farm decision-making [7]. As a result, there is a growing demand for diagnostic tools that are accurate, portable, easy to use, and can provide real-time results [8,9]. Therefore, it is critical to develop rapid, reliable, and field-deployable diagnostic tools that detect SBSEC associated with mastitis while distinguishing them from those used in dairy fermentation.

SBSEC poses a significant challenge for developing field diagnostic methods due to the close phylogenetic relationships among its species, making traditional markers ineffective. Because of their high genetic similarity, these conventional methods often fail to accurately identify individual species [10]. To overcome this limitation, bioinformatic technologies can be utilized to provide a more precise approach to differentiate these closely related species [11]. These technological innovations have enabled the discovery of genetic markers specific to individual species, even within bacterial groups highly similar at the genomic level, such as *Salmonella* and *Listeria monocytogenes* [12,13]. This study leveraged these advancements to analyze the genomes of SBSEC species, aiming to identify genetic markers for each. By employing advanced bioinformatic tools, the genomic sequences of various SBSEC members were thoroughly analyzed to pinpoint conserved genetic regions specific to each species, allowing reliable differentiation from closely related species.

Motivated by these advancements and the need for improved diagnostic tools in the dairy industry, this research focused on SBSEC. Specifically, this study aimed to develop a field-deployable polymerase chain reaction (PCR) method that could accurately and rapidly differentiate between SBSEC species in clinical and industrial settings. Bioinformatic approaches were utilized to analyze the genomes of various SBSEC species, identifying species-specific genetic markers that serve as the foundation for diagnostic assays. This method was designed to bridge the gap between laboratory-based research and practical, on-site applications, offering a valuable tool for disease management and quality assurance in dairy production.

## 2. Materials and Methods

### 2.1. Reference Strain Collection and DNA Extraction

Reference strains were acquired from culture collections (Appendix A) and cultured under standard conditions in brain–heart infusion (BHI, MB cell, Seoul, Republic of Korea) agar at 37 °C for 24 h. Genomic DNA was extracted from cultured cells using a DNeasy blood and tissue kit (Qiagen, Hilden, Germany) according to the manufacturer’s protocol.

### 2.2. Bioinformatic Analysis for Genetic Marker Identification

To identify genetic markers within SBSEC, 206 SBSEC genomes were downloaded from the National Center for Biotechnology Information (NCBI) database (Appendix A). To ensure the accurate classification of the downloaded genomes, average nucleotide identity (ANI) analysis was performed using the ANI calculator [14]. After classification verification, candidate genetic markers were identified using Roary version 3.11.2 with default parameters, including a 95% minimum identity threshold for BLASTP and a core genome threshold of 99%. The candidate genetic markers were subjected to BLAST search analysis against publicly available genomes in the NCBI database to confirm their specificity as unique genetic markers.

### 2.3. Primer Design for On-Site Detection

Primers for on-site detection were designed using the Primer Designer tool version 3.0 (Scientific and Education Software, Durham, NC, USA) based on the identified genetic markers. Design criteria included optimal annealing temperatures, minimal secondary structure formation, and high specificity. In silico validation was conducted by aligning the designed primers against the reference genomes of SBSEC species and the related nontarget species using the Primer-BLAST tool (https://www.ncbi.nlm.nih.gov/tools/primer-blast/) (accessed on 14 April 2024).

### 2.4. Development of Portable Quantitative PCR (qPCR)

The partial regions of genetic markers identified through bioinformatic analysis were amplified using the GenChecker UF-150 real-time PCR system (Genesystem, Daejeon, Republic of Korea; Figure 1A). This compact device with a built-in battery is highly suitable for on-site detection. It features a user-friendly interface (Figure 1A, label 1), a microfluidic chip loading area and thermal cycling unit (Figure 1A, label 2), and a groove for easy handling (Figure 1A, label 3), making the device more portable and convenient to use. To develop a portable qPCR that could detect four species within SBSEC, a microfluidic chip was designed with 10 wells, each dedicated to detecting a specific species or serving as a control. The chip configuration was as follows: wells 1 and 2 for *S. alactolyticus*, wells 3 and 4 for *S. equinus*, wells 5 and 6 for *S. gallolyticus*, wells 7 and 8 for *S. lutetiensis*, and wells 9 and 10 for the negative control (Figure 1B).

Each reaction mixture, consisting of 5 µL of FastFACT 2× qPCR Master Mix Evagreen (Biofact, Daejeon, Republic of Korea), 1 µL of each primer (500 nM), 10 ng target DNA, and sterile distilled water to reach a final volume of 10 µL was loaded into the corresponding well of the microfluidic chip. PCR cycling conditions included denaturation, annealing, and extension (Figure 1C). The cycling program involved an initial preheating step at 95 °C for 1 min, followed by 40 cycles of denaturation at 95 °C for 5 s, annealing at 60 °C for 5 s, and extension at 72 °C for 5 s. Real-time monitoring of the amplification process was performed using GeneRecoder version 3.0.0.6 integrated with the GenChecker system (Figure 1D).

### 2.5. Specificity and Sensitivity Testing of the Developed Portable qPCR Method

Specificity was evaluated by testing against 47 microbial species, including those commonly found in dairy farm environments. To assess sensitivity, serial dilutions of each target SBSEC strain were prepared by culturing the strains and diluting them to concentrations ranging from 10^7^ to 10^0^ colony-forming units (CFU)/mL. DNA was extracted from each dilution and subjected to PCR to determine the lowest detectable concentration of each target species. For sensitivity testing within a food matrix, a cocktail was prepared by mixing SBSEC strains and inoculated into sterile milk to achieve a final concentration ranging from 10^7^ to 10^0^ CFU/mL. Three biological replicates were performed using sterile milk from three different packages. DNA was extracted from each replicate, followed by PCR, to evaluate the ability to detect SBSEC strains in a realistic food matrix, even with potential inhibitors and competing microorganisms. Evaluation of other putative microorganisms in sterile milk was not conducted. All experiments were repeated using three different units of the same instrument model to verify the consistency of the results. The average cycle threshold (Ct) values and standard deviations were calculated to ensure reproducibility and reliability.

### 2.6. SBSEC Isolation and Identification

Raw milk samples were collected from farms in Gimpo, Hwaseong, and Pyeongtaek in Gyeonggi Province, South Korea, placed in sterile containers, and transported to the laboratory under refrigeration. Upon arrival, the samples were immediately inoculated onto BHI (MB cell) agar and incubated at 37 °C for 24 h. After incubation, colonies with typical SBSEC morphology were selected for further identification using the developed portable qPCR and validated against matrix-assisted laser desorption–ionization time-of-flight mass spectrometry (MALDI-TOF MS).

### 2.7. Whole-Genome Sequencing (WGS) for the Identification of Discrepant Isolates

For isolates where species identification differed from MALDI-TOF and portable qPCR results, WGS was conducted to accurately determine their species identity. A total of three isolates were sequenced. Genomic DNA was extracted from these isolates using the method previously described in this study. Libraries were prepared using a TruSeq Nano DNA LT kit (Illumina, San Diego, CA, USA) according to the manufacturer’s protocol. The libraries were sequenced on an Illumina Nextseq (Illumina) to generate paired-end reads with a read length of 300 bp. Raw sequence data were quality-checked using FastQC version 0.11.9 with default parameters. Low-quality bases and adapter sequences were trimmed using Sickle version 1.33 with the default settings, which include a quality threshold of 20 and a minimum length of 20 bp. De novo assembly was performed using SPAdes version 4.0.0 with default parameters, including k-mer sizes of 21, 33, 55, 77, and 99. The genomes were annotated with the Prokaryotic Genome Annotation Pipeline, and species identity was confirmed by comparing them to reference genomes and ANI.

### 2.8. Evaluation of On-Site Applicability

The on-site applicability of the developed portable qPCR was evaluated by conducting field tests at eight dairy farms in Gyeonggi Province (Gimpo, Hwaseong, and Pyeongtaek cities). Sampling interventions were conducted from various sources within each farm, including 33 freshly collected raw milk samples, 20 samples from cow udders, and 1 sample from a cow’s nasal cavity. Environmental samples included 6 from water troughs and 3 from feed troughs. Additionally, commercial dairy products were sampled, including 3 powdered milk samples, 20 other dairy products, and 14 cheese samples. To perform the interventions, pipette swabs (3M Pipette Swab Plus, 3M, St. Paul, MN, USA) were used, and the samples were collected from different surfaces. We established a simple on-site DNA extraction method by modifying the protocol of a direct^TM^ extraction kit (Genesystem) provided by the manufacturer, eliminating the need for additional equipment such as vortexing or centrifugation. Using this modified method, DNA was directly extracted on-site. Immediately after sampling, the samples were placed into a tube containing the extraction buffer (Genesystem), mixed manually, and incubated at room temperature for 5 min. The supernatant, obtained without centrifugation, was directly used for further analysis. This extracted DNA was directly applied to the microfluidic wells of the portable qPCR device to test for SBSEC strains. Real-time monitoring of the amplification process was performed on-site using GeneRecoder (Figure 1D). The specific amplification (Figure 1E) and melting curve analysis (Figure 1F) further confirmed species-specific detection during field testing. The same samples were transported to the laboratory for additional analysis to further validate the accuracy of the portable qPCR results. DNA was reextracted from the samples in the laboratory using a conventional DNA extraction kit (DNeasy blood and tissue kit). The extracted DNA was subjected to a second round of PCR amplification to compare to on-site results.

## 3. Results and Discussion

### 3.1. Comprehensive Analysis of SBSEC Genomes

A comprehensive analysis of 206 SBSEC genomes was performed using bioinformatic tools to evaluate their classification accuracy against existing genomic databases. Each genome met specific quality criteria during the selection process, ensuring reliability in classification and analysis. In the phylogenetic tree (Figure 2A), species relationships were highlighted, with potential misclassifications marked by red squares. These were confirmed by comparing them to species designations in the NCBI database. Although most genomes aligned with their taxonomic classifications, discrepancies were identified, indicating potential mislabeling. For instance, nine genomes labeled *S. equinus* showed only 86.13% to 87.08% ANI identity with *S. equinus* NCTC 12969^T^ (Table 1), below the accepted species-level threshold [15]. However, most genomes exhibited >95% ANI identity with the recently classified *S. ruminicola* CNU_G2^T^, suggesting that they are more closely related to *S. ruminicola*. Two genomes showed <95% ANI with their original and proposed classifications, with one having 94.15% identity to *S. vicugnae* and the other 94.50% to *S. ruminicola*, indicating that they might represent distinct or novel species within *Streptococcus*. Similarly, three genomes originally classified as *S. lutetiensis* had higher ANI identity with *S. ruminicola* CNU_G2^T^ than with *S. lutetiensis* NCTC 13774^T^. However, their ANI values remained <95%, suggesting that they could also represent distinct species, or potentially new ones, within *Streptococcus*, raising the possibility of mislabeling in the NCBI database.

This study emphasizes the importance of comprehensive genomic analyses, such as ANI, for accurate bacterial classification within SBSEC. Misclassified genomes, particularly those labeled as *S. equinus* and *S. lutetiensis*, highlight the need for taxonomic reevaluation as new genomic data become available. Previous studies also reported misclassified genomes in the NCBI database, reinforcing the challenges of maintaining accurate genomic records [16,17]. The significant ANI identity between *S. equinus* genomes and *S. ruminicola* CNU_G2^T^ suggested that these genomes should be reclassified as *S. ruminicola*. Genomes with ANI values < 95% might represent distinct or novel species, reflecting the evolving understanding of bacterial taxonomy.

### 3.2. Pangenome Analysis of 206 SBSEC Genomes

The pangenome analysis of the 206 SBSEC genomes provided insights into genetic diversity and evolutionary relationships within this bacterial group. The phylogenetic tree and presence–absence heatmap (Figure 2B) illustrated strain clustering based on the gene content. The number of conserved genes plateaued as more genomes were added, indicating a stable core genome (Figure 2C), whereas the total number of genes increased, reflecting SBSEC genetic diversity. The rapid growth of new genes with the first few genomes and their stabilization, with a low and stable number of unique genes (Figure 2D), suggested that most variability comes from the accessory genome, with few strain-specific genes.

The analysis identified 24,117 gene clusters, of which only 119 clusters (<1%) constituted the core genome, present in all 206 strains (Figure 2E). Although the core genome represented a small fraction of the total pangenome, these genes formed the foundation for understanding the essential functions conserved across SBSEC species [18]. These core genes likely play a role in fundamental processes necessary for survival in diverse environments [19]. In addition, 222 clusters were found in ≥95% of the strains, and 3772 were present in at least 30 strains. However, most of the pangenome consisted of 20,004 gene clusters in <30 strains, highlighting the extensive genetic diversity within SBSEC. The predominance of gene clusters in a few strains underscored the genetic variability likely driven by adaptations to different environmental niches or host interactions [20].

### 3.3. Identification of Genetic Markers

Genetic markers were identified based on the pangenome analysis by focusing on gene clusters unique to specific SBSEC species. These unique gene clusters were consistently present in all strains of a given species but absent from others, making them ideal candidates for species-specific genetic markers.

In *S. alactolyticus*, the gene encoding 4-diphosphocytidyl-2-C-methyl-D-erythritol kinase was identified as a specific genetic marker. This enzyme plays a crucial role in the non-mevalonate pathway of isoprenoid biosynthesis, which may fulfill the metabolic requirements of *S. alactolyticus* [21]. Similarly, in *S. gallolyticus*, the acetyltransferase gene was identified as a genetic marker. Acetyltransferase is involved in various cellular processes, including modifying proteins and metabolites, and it may be essential for *S. gallolyticus* to thrive in its specific ecological niche [22]. For *S. equinus*, a gene encoding cytokinin riboside 5′-monophosphate phosphoribohydrolase was identified as species-specific. This enzyme is potentially involved in cytokinin metabolism, which may regulate growth and differentiation processes uniquely adapted to *S. equinus* [23]. In *S. lutetiensis*, a hypothetical protein was discovered as a unique marker. Although the function of this protein is still unknown, its consistent presence across *S. lutetiensis* strains suggests that it may have a critical, yet unidentified role in the biology of this species. Although these genes were identified as genetic markers, they were not absent from other species within SBSEC. However, gene sequences in other species differed by >50%, justifying their classification as unique genes for the species in which they were identified.

### 3.4. Specificity Testing

Species-specific primers were designed based on genetic markers identified using pangenome analysis (Table 2). Using these primers, portable qPCR was successfully established to detect genetic markers within SBSEC. This assay was specifically designed for rapid field deployment using a flat plate system, such as the GenChecker UF-150 (Genesystem), which facilitates rapid thermal cycling and real-time detection, completing 40 PCR cycles in 20 min. The compact and portable design of this system further enhances its suitability for environments without traditional laboratory facilities. In contrast, conventional PCR for pathogen detection, such as those used for bovine mastitis, typically requires ≥90 min for 40 cycles and involves heavy, nonportable equipment [24,25]. The advancement of portable qPCR significantly reduces the time and logistical constraints of traditional diagnostic methods, demonstrating the value of innovative engineering features in enabling efficient on-site diagnostics [26].

The specificity of newly developed primers was tested using DNA from 47 bacterial species, including target SBSEC species. Successful amplification was achieved only in target species, with consistent amplification curves (Figure 3). The Ct ranged from 12.97 ± 0.03 to 15.44 ± 0.04 (Appendix A). No amplification occurred in nontarget species, even after 40 PCR cycles, highlighting the robustness of the primers in distinguishing closely related species within SBSEC. The performance of portable qPCR was evaluated using three different devices, with consistent Ct with a standard deviation of <0.05, underscoring its reliability (Appendix A). This high-level reliability is crucial for rapid and accurate species identification in clinical diagnostics, agricultural monitoring, and environmental surveillance. Further validation of assay specificity was achieved through melting curve analysis, confirming the distinctiveness of amplification products. SBSEC species exhibited a specific melting temperature of 80.69 ± 0.19 °C (*S. alactolyticus*), 80.69 ± 0.19 °C (*S. equinus*), 81.72 ± 0.29 °C (*S. gallolyticus*), and 83.77 ± 0.00 °C (*S. lutetiensis*). The single, sharp melting peaks indicated the absence of nonspecific amplification, verifying the high specificity of the primer sets. However, the lack of systematic specificity evaluation across diverse sample matrices represents a limitation of this study.

Establishing this portable qPCR for SBSEC detection significantly advances rapid bacterial identification. Compared to traditional PCR methods, which are time-consuming and require extensive laboratory infrastructure, this assay offers a quick turnaround time, high specificity, and portability. Its consistent performance and specificity suggest that it could be adapted for detecting other pathogens, making it a versatile tool for public health, veterinary medicine, and food safety. Assay reliability across various conditions highlights its potential for broad microbial diagnostics in diverse settings.

### 3.5. Sensitivity Testing

The sensitivity of the primers developed for detecting SBSEC species was evaluated using pure cultures and spiked food samples. Results demonstrated that the primers were highly sensitive, with a limit of detection (LOD) of 10^1^ CFU/mL for most species in pure cultures and spiked food samples, except for *S. alactolyticus*, which exhibited a lower LOD of 10^0^ CFU/mL (Figure 4), indicating that the primers can detect as few as 10^1^ or 10^0^ CFU/mL of target SBSEC species in testing matrices. Standard curves showed high linearity, with R^2^ > 0.98 across 10^7^ to 10^1^ or 10^0^ CFU/mL in pure cultures and spiked food samples (Figure 5). The LOD in this study was superior to the sensitivity reported in previous studies for similar bacterial detection methods. For instance, other studies have reported LODs ranging from 10^2^ to 10^3^ CFU/mL for various bacterial species, such as *Escherichia coli* and *Clostridium difficile*, and serovars using on-site detection methods [27,28]. The lower LOD in this study suggested that the primers developed for SBSEC are particularly effective, potentially offering a more sensitive detection method compared to existing techniques.

The ability to detect such low SBSEC levels in pure culture and complex food matrices underscores the practical applicability of this method in real-world scenarios. In particular, because SBSEC species can cause infections, such as mastitis, in cattle with even low bacterial counts, the high sensitivity of these primers is crucial for on-site diagnostic applications where immediate, enrichment-free detection is necessary. This method ensures the reliable detection of SBSEC species, enhancing overall safety and quality-control processes in food production and clinical environments.

### 3.6. Validation and Comparative Analysis of Portable qPCR for Accurate Diagnostics

To verify the detection accuracy of the developed portable qPCR, raw milk isolates were identified using the newly developed device. Its performance was compared to MALDI-TOF MS, commonly used in microbial diagnostics. During validation, a discrepancy in species identification was observed for a subset of isolates (Table 3). Of the 28 isolates tested, 25 showed consistent results across two methods, but 3 displayed discordant results. For instance, two isolates identified as *S. infantarius* by MALDI-TOF MS were identified as *S. lutetiensis* by the portable qPCR. Similarly, one isolate identified as *S. lutetiensis* by MALDI-TOF MS was identified as *S. gallolyticus* by the portable qPCR. WGS followed by ANI analysis was conducted to resolve these discrepancies, confirming that all three isolates were correctly identified by portable qPCR, aligning with its results rather than those of MALDI-TOF MS (Table 3).

These findings suggest that portable qPCR has superior accuracy in identifying SBSEC species than MALDI-TOF MS. Discrepancies with MALDI-TOF MS can be attributed to their limitations in distinguishing closely related species within the SBSEC group. MALDI-TOF MS, which relies on protein profiling, lacks the resolution necessary for precise species-level identification [29]. The alignment of portable qPCR with WGS underscores its reliability and specificity. This method is a rapid, accurate, and cost-effective alternative to WGS for identifying SBSEC species, eliminating the need for more time-consuming and expensive genomic analyses. The successful validation of this method highlights its potential in food safety testing and public health monitoring, where rapid and accurate pathogen identification is essential.

### 3.7. On-Site Applicability

The field applicability of the developed portable qPCR was evaluated by testing samples directly at dairy farms and from commercial dairy products. DNA extraction, conducted using a direct buffer extraction method, was completed in <5 min after sample swabbing (Figure 6). This rapid extraction method with PCR amplification enabled the entire process, from sample collection to detection, to be completed within 30 min. Portable qPCR detected SBSEC species in 51 of the 100 samples tested (Table 4). *S. equinus* and *S. gallolyticus* were the most commonly identified species, whereas *S. alactolyticus* and *S. lutetiensis* were detected less frequently. *S. equinus* was found in 35 samples, with 33 (94.3%) originating from dairy farms and 2 (5.7%) from dairy products. The lower detection rate in dairy products may be related to previous studies associating *S. equinus* more closely with diseases [30,31]. *S. gallolyticus* was detected in 34 samples, with 25 (73.5%) from dairy farms and 9 (26.5%) from dairy products. Although *S. gallolyticus* is also associated with diseases, it has a connection to dairy product fermentation, particularly in cheese, explaining its higher prevalence in dairy products [5,32].

The detection of SBSEC species by this portable qPCR highlighted its potential as a rapid diagnostic tool for monitoring strains in animal health and dairy fermentation. Early detection of pathogenic bacteria is crucial for reducing the spread of infection and minimizing economic losses in the dairy industry [33,34]. The ability of portable PCR to rapidly identify harmful pathogens at the sample collection site enables immediate decision-making and significantly improves herd health management [35]. Accurately identifying beneficial bacteria is essential in dairy fermentation processes, contributing to the maintenance and improvement of product quality [36]. Integrating sample processing directly with PCR, such as through direct sampling methods, would further enhance the system’s practicality by reducing manual intervention, and this remains a focus for future improvement in on-site diagnostic workflows.

The same samples were analyzed in the laboratory to further verify the accuracy of portable qPCR. DNA was extracted within 1 h using a conventional extraction kit and amplified using portable qPCR. Results from on-site and laboratory-based methods were identical (Table 4), confirming the reliability of portable qPCR. This demonstrated that the on-site method is time-efficient and produces results comparable to traditional laboratory methods. The reliability and efficiency of portable qPCR make it a valuable tool for field applications, especially in environments lacking laboratory facilities, ensuring timely animal health management and product quality in the dairy industry.

## 4. Conclusions

In conclusion, this study developed a rapid and portable qPCR method for detecting and identifying SBSEC species, significantly bettering traditional laboratory-based techniques. Its high specificity, sensitivity, and reliability make it ideal for field applications where laboratory access is limited. Field tests demonstrated its effectiveness, enabling on-site detection of bovine mastitis and milk quality assessment within 30 min. This novel method, highly accurate and consistent with WGS, is a robust solution for rapid bacterial identification in public health, veterinary medicine, and food safety.

## Figures and Tables

**Figure 1 biomolecules-14-01624-f001:**
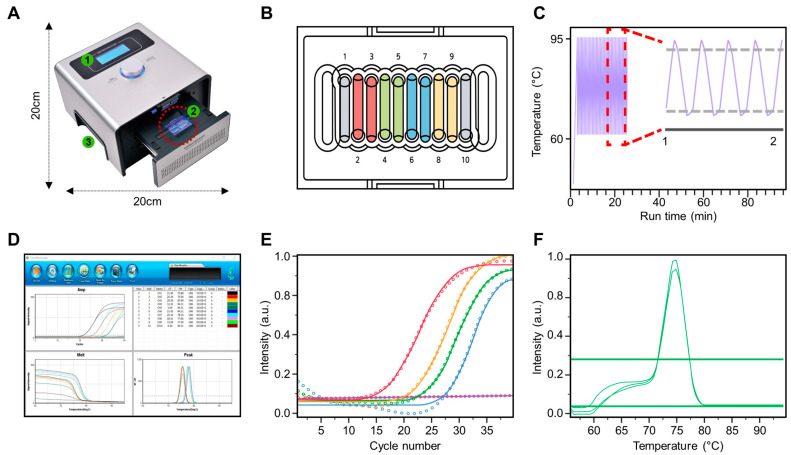
Overview of the portable qPCR system and its components. (**A**) Portable qPCR device, illustrating the components and structure of the system. 1, LCD display; 2, heating plate; 3, groove for easy handling. (**B**) Microfluidic chip used for loading and processing samples in the qPCR device. (**C**) Thermal cycling temperature profiles of the portable qPCR system. (**D**) Real-time amplification screen of the portable qPCR system, displaying amplification progress in different channels. (**E**) Amplification curves generated by the portable qPCR system, representing the increase in fluorescence over cycles for different samples. (**F**) Melting curve analysis, showing the temperature-dependent dissociation of amplified products to assess specificity.

**Figure 2 biomolecules-14-01624-f002:**
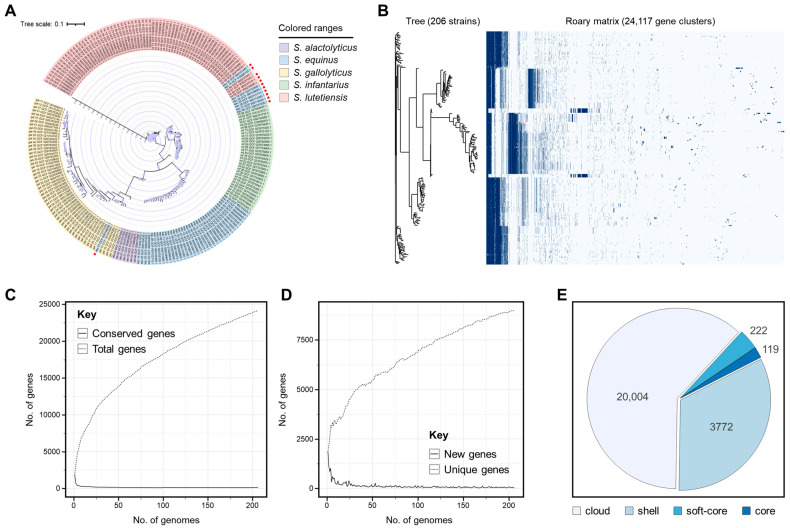
Pangenome analysis of the SBSEC. (**A**) Phylogenetic tree and genome clustering. The circular phylogenetic tree displays the relationships between SBSEC strains based on whole-genome analysis. Each color in the outer ring represents different species within the complex. The tree branches indicate the evolutionary distance between the strains. The red squares in the outer ring highlight genomes that were misclassified. (**B**) The phylogenetic tree on the left shows the relationships among the 206 SBSEC strains, while the right panel displays a gene presence–absence matrix with 24,117 gene clusters. Each row corresponds to a strain and each column to a gene cluster, where dark blue indicates presence and light blue indicates absence. (**C**) This graph illustrates how the number of conserved genes declines as more genomes are analyzed, while the total number of genes continues to rise. (**D**) The plot shows that the number of new genes decreases steeply as additional genomes are added, but the count of unique genes stays nearly constant. (**E**) A pie chart categorizes the pangenome into core genes (99–100% of strains), soft-core genes (95–99%), shell genes (15–95%), and cloud genes (0–15%). The core genome contains 119 genes, the soft core has 222 genes, while the shell and cloud consist of 3772 and 20,004 genes, respectively.

**Figure 3 biomolecules-14-01624-f003:**
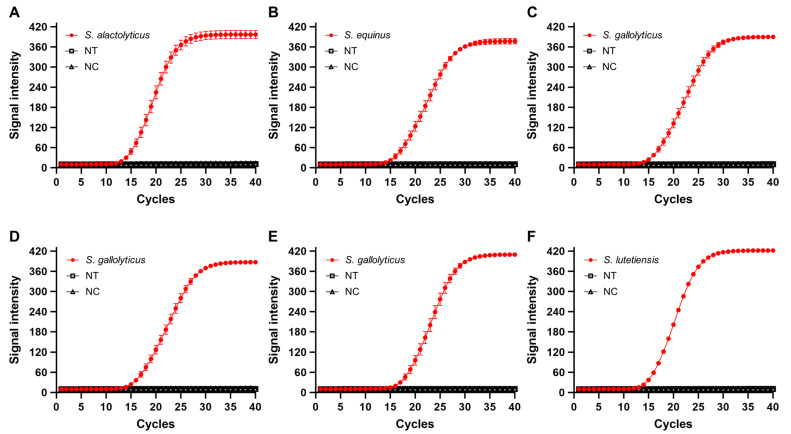
Specificity analysis of portable qPCR for SBSEC species detection. Amplification curves were generated for each target species. (**A**) *S. alactolyticus*, (**B**) *S. equinus*, (**C**) *S. gallolyticus* subsp. *gallolyticus*, (**D**) *S. gallolyticus* subsp. *macedonicus*, (**E**) *S. gallolyticus* subsp. *pasteurianus*, and (**F**) *S. lutetiensis* using species-specific primers. Each qPCR run was performed in triplicate, and the data represent mean values with error bars indicating standard deviations. The absence of amplification in nontarget (NT) and negative control (NC) samples demonstrates the high specificity of the portable qPCR assay for the target species.

**Figure 4 biomolecules-14-01624-f004:**
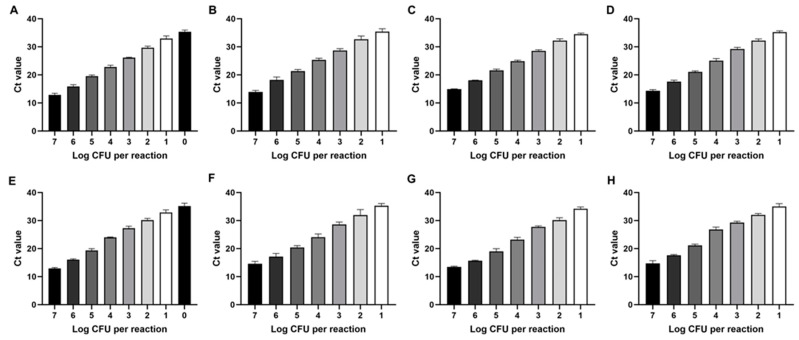
Limit of detection (LOD) for SBSEC using portable qPCR. (**A**) *S. alactolyticus*, (**B**) *S. equinus*, (**C**) *S. gallolyticus*, and (**D**) *S. lutetiensis* in pure culture. (**E**) *S. alactolyticus*, (**F**) *S. equinus*, (**G**) *S. gallolyticus*, and (**H**) *S. lutetiensis* in spiked food samples. Each bar represents qPCR signal intensity for serially diluted pure cultures, ranging from the highest concentration (left) to the lowest concentration (right), showing the sensitivity of detection for each species. The different bar colors represent the logarithmic CFU values (Log CFU per reaction) as indicated on the x-axis, with each color corresponding to a specific concentration step. All tests were conducted in triplicate, with standard deviation bars indicating variability between replicates.

**Figure 5 biomolecules-14-01624-f005:**
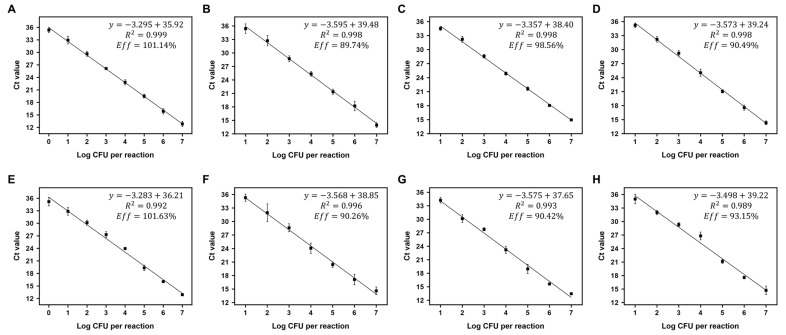
Standard curves for portable qPCR quantification of SBSEC species. Standard curves for SBSEC species detection using portable qPCR, showing the relationship between the logarithm of template concentration and Ct values. Each curve represents tenfold serial dilutions of DNA from the highest to lowest concentrations. (**A**) *S. alactolyticus*, (**B**) *S. equinus*, (**C**) *S. gallolyticus*, and (**D**) *S. lutetiensis* in pure culture. (**E**) *S. alactolyticus*, (**F**) *S. equinus*, (**G**) *S. gallolyticus*, and (**H**) *S. lutetiensis* in spiked food samples. All assays were performed in triplicate, with error bars representing the standard deviation of the replicates.

**Figure 6 biomolecules-14-01624-f006:**
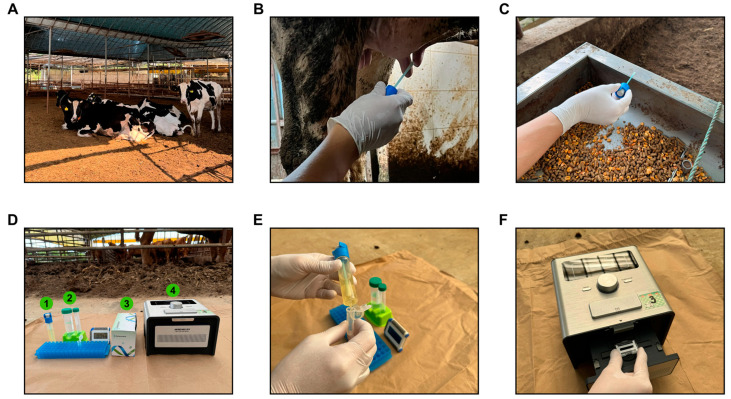
Field diagnostic process of SBSEC using portable qPCR. (**A**) The farm where the samples were collected. (**B**) Swabbing of the udder surface for mastitis diagnosis. (**C**) Environmental sampling from the farm surroundings. (**D**) On-site diagnostic step performed immediately after sampling: 1, collected sample; 2, direct buffer for DNA extraction; 3, on-site diagnostic chip; 4, portable diagnostic device. (**E**) On-site DNA extraction performed immediately after sampling, taking less than 5 min at the farm site. (**F**) On-site analysis using the portable diagnostic device within 20 min.

**Table 1 biomolecules-14-01624-t001:** Comparison of genomes with original and proposed taxa based on ANI values.

Submitted Organisms (Accession No.)	Original Taxa (ANI)	Proposed Taxa (ANI)
*S. equinus* (GCF 900101445.1)	*S. equinus* (86.35%)	*S. ruminicola* (94.38%)
*S. equinus* (GCF 016906125.1)	*S. equinus* (87.08%)	*S. ruminicola* (95.66%)
*S. equinus* (GCF 024732405.1)	*S. equinus* (86.58%)	*S. ruminicola* (95.98%)
*S. equinus* (GCF 900106895.1)	*S. equinus* (86.50%)	*S. ruminicola* (95.89%)
*S. equinus* (GCF 900101715.1)	*S. equinus* (86.43%)	*S. ruminicola* (95.93%)
*S. equinus* (GCF 900112385.1)	*S. equinus* (86.65%)	*S. ruminicola* (95.48%)
*S. equinus* (GCF 900114525.1)	*S. equinus* (86.90%)	*S. ruminicola* (95.62%)
*S. equinus* (GCF 000747205.1)	*S. equinus* (86.92%)	*S. ruminicola* (95.60%)
*S. lutetiensis* (GCF 905237035.1)	*S. lutetiensis* (94.11%)	*S. ruminicola* (94.57%)
*S. lutetiensis* (GCF 905237015.1)	*S. lutetiensis* (93.78%)	*S. ruminicola* (94.50%)
*S. lutetiensis* (GCF 905236995.1)	*S. lutetiensis* (93.78%)	*S. ruminicola* (94.50%)
*S. equinus* (GCF 900104225.1)	*S. equinus* (86.13%)	*S. vicugnae* (94.15%)

**Table 2 biomolecules-14-01624-t002:** Specific primer sequences and amplicon size for detecting SBSEC.

Species	Primer	Sequence (5′-3′)	Size (bp)
*S. alactolyticus*	ala-F	TTG GCA TCT CCA CTC GAA CC	205
	ala-R	TCG GCT CCA CAA CTG AGC AT	
*S. equinus*	equi-F	GGT GGT GAG GTG TAT GGT GTT	108
	equi-R	ACG CTT ACG CTC ATC CAT GT	
*S. gallolyticus*	gallo-F	GGT GTG CCA ATG TCG CTT GA	146
	gallo-R	ACC GCC ATA CGT TGT AGT GTC G	
*S. lutetiensis*	lut-F	TCT CAC GTT GCT AAA GAA AAC CA	148
	lut-R	AAA CCA CTC TTA CAT GAC CGG C	

**Table 3 biomolecules-14-01624-t003:** Comparison of different identification methods (MALDI-TOF MS, portable PCR, and WGS) using isolates from raw milk.

Strain	Source	MALDI-TOF (Score) ^1^	Portable PCR	WGS (ANI %) ^2^
S1	Raw milk	*S. gallolyticus* (2.72)	*S. gallolyticus*	NA
S2	Raw milk	*S. gallolyticus* (2.23)	*S. gallolyticus*	NA
S3	Raw milk	*S. gallolyticus* (2.25)	*S. gallolyticus*	NA
S4	Raw milk	*S. gallolyticus* (2.06)	*S. gallolyticus*	NA
S5	Raw milk	*S. gallolyticus* (2.09)	*S. gallolyticus*	NA
S6	Raw milk	*S. lutetiensis* (2.10)	*S. lutetiensis*	NA
S7	Raw milk	*S. lutetiensis* (2.07)	*S. gallolyticus*	*S. gallolyticus* (96.5%)
S8	Raw milk	*S. lutetiensis* (2.00)	*S. lutetiensis*	NA
S9	Raw milk	*S. infantarius* (2.09)	*S. lutetiensis*	*S. lutetiensis* (99.2%)
S10	Raw milk	*S. infantarius* (2.25)	*S. lutetiensis*	*S. lutetiensis* (99.4%)
S11	Raw milk	*S. equinus* (2.25)	*S. equinus*	NA
S12	Raw milk	*S. equinus* (2.08)	*S. equinus*	NA
S13	Raw milk	*S. equinus* (2.24)	*S. equinus*	NA
S14	Raw milk	*S. gallolyticus* (1.97)	*S. gallolyticus*	NA
S15	Raw milk	*S. gallolyticus* (2.20)	*S. gallolyticus*	NA
S16	Raw milk	*S. gallolyticus* (2.19)	*S. gallolyticus*	NA
S17	Raw milk	*S. gallolyticus* (2.20)	*S. gallolyticus*	NA
S18	Raw milk	*S. gallolyticus* (2.25)	*S. gallolyticus*	NA
S19	Raw milk	*S. gallolyticus* (2.18)	*S. gallolyticus*	NA
S20	Raw milk	*S. gallolyticus* (2.22)	*S. gallolyticus*	NA
S21	Raw milk	*S. gallolyticus* (2.33)	*S. gallolyticus*	NA
S22	Raw milk	*S. gallolyticus* (2.25)	*S. gallolyticus*	NA
S23	Raw milk	*S. gallolyticus* (2.27)	*S. gallolyticus*	NA
S24	Raw milk	*S. gallolyticus* (2.33)	*S. gallolyticus*	NA
S25	Raw milk	*S. gallolyticus* (2.30)	*S. gallolyticus*	NA
S26	Raw milk	*S. gallolyticus* (2.32)	*S. gallolyticus*	NA
S27	Raw milk	*S. gallolyticus* (2.28)	*S. gallolyticus*	NA
S28	Raw milk	*S. gallolyticus* (2.21)	*S. gallolyticus*	NA

^1^ MALDI-TOF MS scores of ≥2.0 were accepted for species assignment and scores of ≥1.7 but <2.0 for identification to the genus level. ^2^ Average nucleotide identity (ANI) values of ≥95% indicate species-level identification.

**Table 4 biomolecules-14-01624-t004:** Monitoring results of portable qPCR for environmental and milk-related products across various farms.

Source	Sampling Location	Amplification of Portable qPCR (Direct Buffer/Kit) ^1,2^			
		SA	SE	SG	SL
Raw milk	Farm A, South Korea	−/−	−/−	+/+	−/−
Raw milk	Farm A, South Korea	−/−	−/−	−/−	−/−
Raw milk	Farm A, South Korea	−/−	+/+	−/−	+/+
Raw milk	Farm A, South Korea	−/−	+/+	−/−	−/−
Raw milk	Farm A, South Korea	−/−	+/+	−/−	+/+
Raw milk	Farm B, South Korea	−/−	+/+	−/−	−/−
Raw milk	Farm B, South Korea	−/−	+/+	−/−	−/−
Raw milk	Farm B, South Korea	−/−	−/−	−/−	−/−
Raw milk	Farm B, South Korea	−/−	−/−	−/−	−/−
Raw milk	Farm B, South Korea	−/−	+/+	−/−	−/−
Raw milk	Farm C, South Korea	−/−	−/−	−/−	−/−
Raw milk	Farm C, South Korea	−/−	−/−	−/−	−/−
Raw milk	Farm C, South Korea	−/−	+/+	−/−	−/−
Raw milk	Farm C, South Korea	−/−	+/+	−/−	−/−
Raw milk	Farm C, South Korea	−/−	+/+	−/−	−/−
Raw milk	Farm D, South Korea	−/−	+/+	+/+	−/−
Raw milk	Farm D, South Korea	−/−	+/+	−/−	−/−
Raw milk	Farm D, South Korea	−/−	+/+	−/−	−/−
Raw milk	Farm D, South Korea	−/−	+/+	−/−	−/−
Raw milk	Farm D, South Korea	−/−	+/+	−/−	−/−
Raw milk	Farm E, South Korea	−/−	−/−	−/−	−/−
Raw milk	Farm E, South Korea	−/−	−/−	−/−	−/−
Raw milk	Farm E, South Korea	−/−	+/+	−/−	−/−
Raw milk	Farm E, South Korea	−/−	−/−	−/−	−/−
Raw milk	Farm F, South Korea	−/−	−/−	−/−	−/−
Raw milk	Farm F, South Korea	−/−	−/−	−/−	−/−
Raw milk	Farm F, South Korea	−/−	−/−	−/−	−/−
Raw milk	Farm G, South Korea	−/−	−/−	−/−	−/−
Raw milk	Farm G, South Korea	−/−	−/−	−/−	−/−
Raw milk	Farm G, South Korea	−/−	−/−	−/−	−/−
Raw milk	Farm H, South Korea	−/−	−/−	−/−	−/−
Raw milk	Farm H, South Korea	−/−	−/−	−/−	−/−
Raw milk	Farm H, South Korea	−/−	−/−	−/−	−/−
Cow’s udder	Farm A, South Korea	−/−	+/+	+/+	+/+
Cow’s udder	Farm A, South Korea	−/−	+/+	+/+	−/−
Cow’s udder	Farm A, South Korea	−/−	+/+	+/+	−/−
Cow’s udder	Farm A, South Korea	−/−	+/+	+/+	−/−
Cow’s udder	Farm A, South Korea	−/−	+/+	+/+	+/+
Cow’s udder	Farm B, South Korea	−/−	+/+	+/+	+/+
Cow’s udder	Farm B, South Korea	−/−	+/+	−/−	+/+
Cow’s udder	Farm B, South Korea	−/−	+/+	−/−	−/−
Cow’s udder	Farm B, South Korea	−/−	−/−	+/+	−/−
Cow’s udder	Farm B, South Korea	−/−	+/+	+/+	−/−
Cow’s udder	Farm C, South Korea	−/−	+/+	+/+	−/−
Cow’s udder	Farm C, South Korea	−/−	+/+	−/−	−/−
Cow’s udder	Farm C, South Korea	−/−	+/+	+/+	−/−
Cow’s udder	Farm C, South Korea	−/−	+/+	+/+	−/−
Cow’s udder	Farm C, South Korea	−/−	+/+	+/+	−/−
Cow’s udder	Farm D, South Korea	−/−	+/+	+/+	−/−
Cow’s udder	Farm D, South Korea	−/−	−/−	−/−	−/−
Cow’s udder	Farm D, South Korea	−/−	−/−	−/−	−/−
Cow’s udder	Farm D, South Korea	−/−	−/−	−/−	+/+
Cow’s udder	Farm D, South Korea	−/−	−/−	−/−	−/−
Water trough	Farm A, South Korea	−/−	−/−	+/+	−/−
Water trough	Farm A, South Korea	−/−	−/−	−/−	−/−
Water trough	Farm B, South Korea	−/−	−/−	−/−	−/−
Water trough	Farm C, South Korea	−/−	−/−	−/−	−/−
Water trough	Farm D, South Korea	−/−	−/−	−/−	−/−
Water trough	Farm D, South Korea	−/−	−/−	−/−	−/−
Cow’s nasal	Farm A, South Korea	−/−	−/−	−/−	−/−
Feed trough	Farm A, South Korea	−/−	−/−	−/−	−/−
Feed trough	Farm B, South Korea	−/−	+/+	−/−	−/−
Feed trough	Farm C, South Korea	−/−	−/−	−/−	−/−
Powdered milk	India	−/−	−/−	−/−	−/−
Powdered milk	India	−/−	−/−	+/+	−/−
Powdered milk	India	−/−	−/−	−/−	−/−
Dairy product	South Korea	−/−	−/−	−/−	−/−
Dairy product	South Korea	−/−	−/−	−/−	−/−
Dairy product	South Korea	+/+	−/−	−/−	−/−
Dairy product	South Korea	−/−	−/−	+/+	−/−
Dairy product	South Korea	−/−	−/−	−/−	−/−
Dairy product	South Korea	−/−	−/−	−/−	−/−
Dairy product	South Korea	−/−	−/−	−/−	−/−
Dairy product	South Korea	−/−	−/−	−/−	−/−
Dairy product	South Korea	−/−	−/−	−/−	−/−
Dairy product	South Korea	−/−	−/−	−/−	−/−
Dairy product	South Korea	−/−	−/−	−/−	−/−
Dairy product	South Korea	−/−	−/−	−/−	−/−
Dairy product	South Korea	−/−	−/−	−/−	−/−
Dairy product	South Korea	−/−	−/−	−/−	−/−
Dairy product	South Korea	−/−	−/−	−/−	−/−
Dairy product	South Korea	−/−	−/−	−/−	−/−
Dairy product	South Korea	−/−	−/−	−/−	−/−
Dairy product	South Korea	−/−	−/−	−/−	−/−
Dairy product	South Korea	−/−	−/−	−/−	−/−
Dairy product	South Korea	−/−	−/−	−/−	−/−
Cheese	Italy	+/+	−/−	−/−	−/−
Cheese	Greece	−/−	−/−	+/+	−/−
Cheese	Greece	−/−	−/−	+/+	−/−
Cheese	Italy	−/−	−/−	−/−	−/−
Cheese	Italy	−/−	−/−	+/+	−/−
Cheese	Italy	−/−	−/−	+/+	+/+
Cheese	Italy	−/−	−/−	−/−	−/−
Cheese	Italy	−/−	−/−	−/−	−/−
Cheese	South Korea	−/−	−/−	−/−	−/−
Cheese	South Korea	−/−	−/−	+/+	−/−
Cheese	Spain	−/−	−/−	+/+	−/−
Cheese	France	−/−	+/+	−/−	−/−
Cheese	Italy	−/−	−/−	−/−	+/+
Cheese	Italy	−/−	+/+	+/+	−/−

^1^ +, Presence of species; −, absence of species. The first symbol represents the result using the direct buffer (on-site method), and the second symbol represents the result using the DNA extraction kit (laboratory method). ^2^ SA, *S. alactolyticus*; SE, *S. equinus*; SG, *S. gallolyticus*; SL, *S. lutetiensis*.

## Data Availability

Data are contained in the article and Appendix A.

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
