# Peer review of "Direct On-Chip Diagnostics of Streptococcus bovis/Streptococcus equinus Complex in Bovine Mastitis Using Bioinformatics-Driven Portable qPCR"

_biomolecules, 2024, doi:10.3390/biom14121624_

Round 1

Reviewer 1 Report

Comments and Suggestions for Authors

The study presents a novel portable qPCR diagnostic tool for detecting Streptococcus bovis/Streptococcus equinus complex (SBSEC). This study represents an advance in rapid bacterial diagnostics. However, some issues need to be addressed, mainly in the Materials and Methods section:

-          - Please, describe the Roary default parameters (lines 91-92)

-         -  The specificity was evaluated by testing against 47 microbial species. However, robustness testing with varying sample matrices and different SBSEC strain mixes could have been tested. Have the authors taken this aspect into account?.

-          - Authors should describe how they performed the serial dilutions of each target SBSEC strain to assess sensitivity. Furthermore, authors state that for sensitivity testing within a food matrix, a cocktail was prepared by mixing SBSEC strains and inoculated into sterile milk to achieve a final concentration. After this, PCR was tested on these milk-based samples to evaluate its ability to detect SBSEC strains in the food matrix, even with potential inhibitors and competing microorganisms commonly found in dairy products. However, have the authors evaluated the other putative microorganisms in a sterile milk? And how many replicates were performed for this (three times ????), have the authors used the same samples???? If the authors have used three replicates of the same milk-based  product, It is difficult to assess reliable results. The authors need to describe the "Specificity and Sensitivity Testing" section in greater depth, as in its current form it includes few replicates, and the results might be unreliable.

-        -   In Whole-genome sequencing (WGS) for the identification of discrepant isolates, authors should indicate the number of isolates sequenced. Indicate software for quality analysis of raw sequences (FastQC). If the authors have processed the raw reads for trimming low-quality bases and removing adapters, using tools such as Trimmomatic or Cutadapt, this should be indicated. In general, the authors should indicate the parameters used in the different software tools, such as SPAdes and others.

-          - In Evaluation of on-site applicability section, authors should indicate the number of samples from the different sources within each farm  analysed (farm environment, the udders of dairy cows, freshly collected raw milk, and commercial dairy products).

In Results and discussion section:

In lines 188-189, authors indicate that nine genomes labeled S. equinus showed only 86.13% to 87.08% ANI similarity with S. equinus NCTC 12969T (Table 1)…. If the authors refer to exact matches between sequences, it is better to use "identity" instead of "similarity." The same applies throughout the manuscript.

Gene names and symbols should be written in italics.

Author Response

Reviewer #1: The study presents a novel portable qPCR diagnostic tool for detecting Streptococcus bovis/Streptococcus equinus complex (SBSEC). This study represents an advance in rapid bacterial diagnostics. However, some issues need to be addressed, mainly in the Materials and Methods section:

- Please, describe the Roary default parameters (lines 91-92)

Response: As you recommended, we added the Roary default parameters as follows:

Lines 92-94: Roary version 3.11.2 with default parameters, including a 95% minimum identity threshold for BLASTP and a core genome threshold of 99%.

- The specificity was evaluated by testing against 47 microbial species. However, robustness testing with varying sample matrices and different SBSEC strain mixes could have been tested. Have the authors taken this aspect into account?

Response: While specificity testing against varying sample matrices was not directly evaluated in this study, the robustness of SBSEC detection was assessed through monitoring multiple sample types. These tests confirmed that the portable qPCR method could selectively amplify SBSEC even in the presence of other microorganisms commonly found in such environments. However, we acknowledge the limitation of not performing comprehensive specificity tests across a wider range of sample matrices. As you recommended, we added the sentence as follows:

Lines 311-312: However, the lack of systematic specificity evaluation across diverse sample matrices represents a limitation of this study.

- Authors should describe how they performed the serial dilutions of each target SBSEC strain to assess sensitivity. Furthermore, authors state that for sensitivity testing within a food matrix, a cocktail was prepared by mixing SBSEC strains and inoculated into sterile milk to achieve a final concentration. After this, PCR was tested on these milk-based samples to evaluate its ability to detect SBSEC strains in the food matrix, even with potential inhibitors and competing microorganisms commonly found in dairy products. However, have the authors evaluated the other putative microorganisms in a sterile milk? And how many replicates were performed for this (three times ????), have the authors used the same samples???? If the authors have used three replicates of the same milk-based product, It is difficult to assess reliable results. The authors need to describe the "Specificity and Sensitivity Testing" section in greater depth, as in its current form it includes few replicates, and the results might be unreliable.

Response: To assess sensitivity, serial dilutions of each target SBSEC strain were prepared by culturing the strains and diluting them to concentrations ranging from 107 to 100 colony-forming units (CFU)/ml. DNA was extracted from each dilution and used for sensitivity evaluation. The evaluation of other putative microorganisms in sterile milk was not conducted. It is presumed that microorganisms such as Lactobacillus delbrueckii, commonly present in milk, might exist in the matrix. However, assessing these microorganisms would require a metagenomics approach, which was beyond the scope of this study. For sensitivity testing within a food matrix, three biological replicates were performed. Specifically, SBSEC strain cocktails were inoculated into three separate sterile milk units from different packages. DNA was extracted from each replicate, followed by PCR to ensure reliability and reproducibility of the results. As you recommended, we added or revised the sentence as follows:

Lines 135-148: To assess sensitivity, serial dilutions of each target SBSEC strain were prepared by culturing the strains and diluting them to concentrations ranging from 107 to 100 colony-forming units (CFU)/ml. DNA was extracted from each dilution and subjected to PCR to determine the lowest detectable concentration of each target species. For sensitivity testing within a food matrix, a cocktail was prepared by mixing SBSEC strains and inoculated into sterile milk to achieve a final concentration ranging from 107 to 100 CFU/ml. Three biological replicates were performed using sterile milk from three different packages. DNA was extracted from each replicate, followed by PCR, to evaluate the ability to detect SBSEC strains in a realistic food matrix, even with potential inhibitors and competing microorganisms. The evaluation of other putative microorganisms in sterile milk was not conducted. All experiments were repeated using three different units of the same instrument model to verify the consistency of the results. The average cycle threshold (Ct) values and standard deviations were calculated to ensure reproducibility and reliability.

- In Whole-genome sequencing (WGS) for the identification of discrepant isolates, authors should indicate the number of isolates sequenced. Indicate software for quality analysis of raw sequences (FastQC). If the authors have processed the raw reads for trimming low-quality bases and removing adapters, using tools such as Trimmomatic or Cutadapt, this should be indicated. In general, the authors should indicate the parameters used in the different software tools, such as SPAdes and others.

Response: As you recommended, we added the sentence as follows:

Lines 159-160: A total of three isolates were sequenced.

Lines 164-169: Raw sequence data were quality-checked using FastQC version 0.11.9 with default parameters. Low-quality bases and adapter sequences were trimmed using Sickle version 1.33 with the default settings, which include a quality threshold of 20 and a minimum length of 20 bp. De novo assembly was performed using SPAdes version 4.0.0 with default parameters, including the k-mer sizes of 21, 33, 55, 77, and 99.

- In Evaluation of on-site applicability section, authors should indicate the number of samples from the different sources within each farm analysed (farm environment, the udders of dairy cows, freshly collected raw milk, and commercial dairy products).

Response: As you recommended, we added the sentence as follows:

Lines 174-178: Sampling interventions were conducted from various sources within each farm, including 33 freshly collected raw milk samples, 20 samples from cow udders, and 1 sample from a cow's nasal cavity. Environmental samples included 6 from water troughs and 3 from feed troughs. Additionally, commercial dairy products were sampled, including 3 powdered milk samples, 20 other dairy products, and 14 cheese samples.

In Results and discussion section:

- In lines 188-189, authors indicate that nine genomes labeled S. equinus showed only 86.13% to 87.08% ANI similarity with S. equinus NCTC 12969T (Table 1)…. If the authors refer to exact matches between sequences, it is better to use "identity" instead of "similarity." The same applies throughout the manuscript.

Response: As you recommended, we changed “similarity” to “identity” throughout the manuscript.

Lines 206, 207, 210, 212, and 239: identity

- Gene names and symbols should be written in italics.

Response: As you recommended, we revised gene names and symbols in italics.

Line 268: 4-diphosphocytidyl-2-C-methyl-D-erythritol kinase

Line 271: acetyltransferase

Lines 274-275: cytokinin riboside 5¢-monophosphate phosphoribohydrolase

Reviewer 2 Report

Comments and Suggestions for Authors

The rapid (within 30min) and portable/light qPCR method for identifying SBSEC species for on-site detection of bovine mastitis and milk quality assessment within 30 min. It is highly accurate and consistent with WGS, more s-specific then MALDI TOF with significant applicability for robust bacterial identification in public health, veterinary medicine, and food safety. It is a beautiful example of applicability!

I did have a problem of finding more scientific reference for nucleic acid extraction/s. Finally I find it online and realize that there is 5min manual work requiring centrifugation (?) as presented on www link and not presented on Fig 6.

I suggest that authors share this info (if correct?) related to Genesystem N.A. kit and perhaps indicate that next step in improving applicability – should be integration of sample processing with PCR (?) i.e. direct sampling?

https://www.genesystem.co.kr/en/platforms/consumables/extraction_kit

Otherwise I think that the work is excellent but would profit more from sharing more engineering characteristics of GenChecker UF-150 Real-time PCR system (Genesystem, Daejeon, Korea; Figure 1A), or at list references where additional info can be found (patent, MS, or similar), where not only biologist and biochemist but also engineers can be included as potential end-users.

Author Response

Reviewer #2: The rapid (within 30min) and portable/light qPCR method for identifying SBSEC species for on-site detection of bovine mastitis and milk quality assessment within 30 min. It is highly accurate and consistent with WGS, more s-specific then MALDI TOF with significant applicability for robust bacterial identification in public health, veterinary medicine, and food safety. It is a beautiful example of applicability!

I did have a problem of finding more scientific reference for nucleic acid extraction/s. Finally I find it online and realize that there is 5min manual work requiring centrifugation (?) as presented on www link and not presented on Fig 6.

Response: We established a simple on-site DNA extraction method by modifying the protocol of the directTM extraction kit (Genesystem) provided by the manufacturer, eliminating the need for additional equipment such as vortexing or centrifugation. Using this modified method, DNA was directly extracted on-site. As you recommended, we added a detailed explanation of this method.

Lines 180-188: We established a simple on-site DNA extraction method by modifying the protocol of the directTM extraction kit (Genesystem) provided by the manufacturer, eliminating the need for additional equipment such as vortexing or centrifugation. Using this modified method, DNA was directly extracted on-site. Immediately after sampling, the samples were placed into a tube containing the extraction buffer (Genesystem), mixed manually, and incubated at room temperature for 5 min. The supernatant, obtained without centrifugation, was directly used for further analysis. This extracted DNA was directly applied to the microfluidic wells of the portable qPCR device to test for SBSEC strains.

I suggest that authors share this info (if correct?) related to Genesystem N.A. kit and perhaps indicate that next step in improving applicability – should be integration of sample processing with PCR (?) i.e. direct sampling?
https://www.genesystem.co.kr/en/platforms/consumables/extraction_kit

Response: We established a simple on-site DNA extraction method by modifying the protocol of the directTM extraction kit (Genesystem) provided by the manufacturer, eliminating the need for additional equipment such as vortexing or centrifugation. Using this modified method, DNA was directly extracted on-site. As you recommended, we added a detailed explanation of this method. Additionally, we agree that integrating sample processing with PCR, such as implementing direct sampling methods, would be a valuable improvement to enhance the system’s practicality and reduce manual intervention. We suggested this as a future direction for improving on-site diagnostic workflows in the results and discussion section.

Lines 180-188: We established a simple on-site DNA extraction method by modifying the protocol of the directTM extraction kit (Genesystem) provided by the manufacturer, eliminating the need for additional equipment such as vortexing or centrifugation. Using this modified method, DNA was directly extracted on-site. Immediately after sampling, the samples were placed into a tube containing the extraction buffer (Genesystem), mixed manually, and incubated at room temperature for 5 min. The supernatant, obtained without centrifugation, was directly used for further analysis. This extracted DNA was directly applied to the microfluidic wells of the portable qPCR device to test for SBSEC strains.

Lines 424-427: Integrating sample processing directly with PCR, such as through direct sampling methods, would further enhance the system’s practicality by reducing manual intervention, and this remains a focus for future improvement in on-site diagnostic workflows.

Otherwise I think that the work is excellent but would profit more from sharing more engineering characteristics of GenChecker UF-150 Real-time PCR system (Genesystem, Daejeon, Korea; Figure 1A), or at list references where additional info can be found (patent, MS, or similar), where not only biologist and biochemist but also engineers can be included as potential end-users.

Response: As you recommended, we expanded the description of the system, highlighting its key features such as portability, rapid thermal cycling, and real-time detection capabilities. Additionally, we provided references to the manufacturer’s technical documentation in the revised manuscript.

Lines 288-291: such as the GenChecker UF-150 (Genesystem), which facilitates rapid thermal cycling and real-time detection, completing 40 PCR cycles in 20 min. The compact and portable design of this system further enhances its suitability for environments without traditional laboratory facilities.

Lines 293-296: The advancement of portable qPCR significantly reduces the time and logistical constraints of traditional diagnostic methods, demonstrating the value of innovative engineering features in enabling efficient on-site diagnostics [26].

Lines 526-527: Victory Scientific. Operating manual for model UF-150 GENECHECKER® ultra-fast real-time PCR system. Available online: https://victoryscientific.com (accessed on 6 December 2024).

Reviewer 3 Report

Comments and Suggestions for Authors

The paper submitted by Jaewook Kim et al addressed the development of new PCR method to detect some pathogen in bovine milk.

It is a new portable and affidable method to detect Streptococcus bovis/equinus, that is a relevant problem on livestok health. But it is a very powerful prototype to develop new similar method for other pathogens.

The genomic approach to the design of the method is the major novelty. I think it is a good job and it is not necessary any improvement.

The conclusion are well described and congruent with the goal of the job.   The references are appropriate.

Author Response

Reviewer #3: The paper submitted by Jaewook Kim et al addressed the development of new PCR method to detect some pathogen in bovine milk. It is a new portable and affidable method to detect Streptococcus bovis/equinus, that is a relevant problem on livestok health. But it is a very powerful prototype to develop new similar method for other pathogens. The genomic approach to the design of the method is the major novelty. I think it is a good job and it is not necessary any improvement. The conclusion are well described and congruent with the goal of the job. The references are appropriate.

Response: We sincerely thank the reviewer for the positive feedback and valuable comments on our manuscript. We have no further revisions to propose based on these comments and are delighted that the work meets the reviewer’s expectations.

Round 2

Reviewer 1 Report

Comments and Suggestions for Authors

The authors have adequately addressed the aspects suggested by this reviewer. Many thanks to the authors for their effort. I believe the article has improved and can be published in its current form.